# Added and Free Sugars Intake and Metabolic Biomarkers in Japanese Adolescents

**DOI:** 10.3390/nu12072046

**Published:** 2020-07-09

**Authors:** Masayuki Okuda, Aya Fujiwara, Satoshi Sasaki

**Affiliations:** 1Graduate School of Sciences and Technology for Innovation, Yamaguchi University, 1-1-1 Minami-Kogushi, Ube 755-8505, Japan; 2Department of Nutritional Epidemiology and Shokuiku, National Institute of Biomedical Innovation, Health and Nutrition, 1-23-1 Toyama, Shinjuku-ku, Tokyo 162-8636, Japan; fujiwaraay@nibiohn.go.jp; 3Department of Social and Preventive Epidemiology, Graduate School of Medicine, and School of Public Health, The University of Tokyo, 7-3-1 Hongo, Bunkyo-ku, Tokyo 113-0033, Japan; stssasak@m.u-tokyo.ac.jp

**Keywords:** added sugars, adolescents, cardiovascular risks, free sugars, Japanese, metabolic syndrome

## Abstract

Reduction in the intakes of added and free sugars is a recommendation to lower cardiometabolic risks. Sugars intake is considered lowest in the Asian-Pacific region, particularly Japan. We examined the association between sugars intake and cardiometabolic risks in Japanese adolescents. We included 3242 students (mean age, 13.56 years) living in Shunan City, Japan, between 2006 and 2010. Sugars intake was estimated using the brief-type self-administered diet history questionnaire. Anthropometrics, serum lipids, fasting plasma glucose, and blood pressure were measured. Metabolic syndrome was determined by the combination of overweight and other risks. Intakes of added and free sugars were 7.6–7.9%E and 8.4–8.8%E of the total energy intake (%E), respectively. Categories based on quintiles of added or free sugars intakes were associated with fasting glucose, systolic blood pressure, and the z-score of metabolic syndrome (*P_trend_* ≤ 0.025). Other than the association between added sugars ≥10%E and high glucose (odds ratio 1.51, 95% confidence interval 1.04–2.19, *p* = 0.031), non-significantly high intakes of added or free sugars for risks occurred. Association was observed between added or free sugars intake and cardiometabolic biomarkers in Japanese adolescents, and added sugars intake <10%E could prevent glucose intolerance but not metabolic syndrome.

## 1. Introduction

Excess intake of sugars is associated with increased prevalence of obesity [1], cardiometabolic risks [2], and dental caries [3]. Free sugars are defined by the World Health Organization (WHO)/Food and Agriculture Organization (FAO) as all monosaccharides and disaccharides added to foods by manufacturers, cooks or customers, and sugars naturally present in honey, syrups, and fruit juices [4]. The WHO recommends intake of free sugars < 10% of the total energy intake (TEI) and advises the reduction of free sugars to <5% of TEI [4]. Added sugars are defined as sugar and syrups added to food during processing and preparation, excluding sugars that occur naturally in foods [5]. The 2015 US Dietary Guidelines recommend keeping added sugars intake to <10% of TEI [6]. The American Heart Association recommends that added sugars in children be limited to ≤25 g/day [7], which corresponds to 5% of TEI for those who consume 2000 kcal/day.

Obesity and related cardiovascular diseases are a public health burden in Japan as well as in American and European countries [8]. However, among the member countries of the Organization for Economic Co-operation Development (OECD), Japan has the lowest prevalence of obesity [9]. Moreover, recently, Japan has not witnessed an increase in the prevalence of diabetes and prediabetes [10], and the prevalence of childhood obesity has decreased since 2002 [11]. Asians are likely to suffer from diabetes at a relatively low body mass index (BMI) or waist circumference in comparison to Europids [12,13,14], and fat distribution varies among Asian countries [15]. It should be clarified whether added or free sugars could pose additional cardiometabolic risks for Japanese people. 

Japan has the lowest sugar consumption per capita among the developed OECD countries [16]. In the US and UK, the main source of added sugars, especially for adolescents, includes liquid sugars such as sugar-sweetened beverages (SSBs), and sports drinks [17,18]. However, consumption of SSBs in the Asia-Pacific region is the lowest in the world [19]. Further, Japanese use sugar as a seasoning agent in cooking modern Japanese cuisine “washoku” [20]. 

There have been few studies investigating the association between intake of sugars or SSBs and cardiovascular risks among Asian adolescents. Studies on adolescents from Korea [21], China [22], Japan, and Cambodia [23] examined the association of sugars intake with body fatness, while studies from Korea [24] and Iran [25] examined the association with metabolic risk factors, and showed no significant associations. Therefore, the aim of this study was to examine the association between added or free sugars intake and metabolic biomarkers other than adiposity in Japanese adolescents and to evaluate high intake of sugars to identify adolescents with cardiometabolic risks. In meta-analyses on energy-control trials mostly from Europe and the USA [1,26], observed effects of sugars intake on body weight, blood pressure (BP), and blood lipid profile are considered via an excess of energy intake in conjunction with an excess of sugars intake; so we examined the associations with adjustment for TEI.

## 2. Materials and Methods 

### 2.1. Subjects 

This study was part of the Shunan Child Cohort Study described elsewhere in detail [27,28]. Data of the 8th graders attending 17 secondary schools in Shunan City, Japan, between 2006 and 2010 were used. Students who assented and whose guardians provided written informed consent were asked to complete questionnaires, take a blood test, and have anthropometrics and BP measured. This study protocol was in accordance with the Declaration of Helsinki and approved by the Ethics Committee of Yamaguchi University Hospital (H17-14 on 18 May, 2005, H17-14-2 on 22 March, 2006, H22-158 on 26 January, 2011, and H22-158- [1] on 22 March, 2017) and the education board of Shunan City. 

### 2.2. Dietary Assessment

Dietary intake of the students during the last month was assessed using the brief type of self-administered diet history questionnaire for youths (BDHQ15y), which was validated with biomarkers such as carotenoids, tocopherols, fatty acids, and urea nitrogen [29,30]. Nutrition intake estimated from the BDHQ for adults has been validated in comparison with those estimated from dietary records [31]. Sugars intake was estimated using a food composition database developed for Japanese sugars intake [32]. Estimation of added and free sugars from the BDHQ for adults was validated in comparison with the 16-day dietary records (Spearman’s correlation coefficients, 0.40–0.57) [33]. Intakes of sugars, fat, and total dietary fiber were expressed as energy density (percentage of TEI, %E, or g/1000 kcal); fat and total dietary fiber intake has been described elsewhere [28].

### 2.3. Blood Tests

The students were requested to restrict intake of any foods or beverages from 10 p.m. on the day before blood drawing, which was performed after 9 a.m. at school. Whether the students took breakfast or not before blood drawing was recorded. Levels of low-density lipoprotein cholesterol (LDL), high-density lipoprotein cholesterol (HDL), triglyceride (TG), and plasma glucose (Glu) were analyzed at the clinical laboratory of Tokuyama Medical Association Hospital. 

### 2.4. Anthropometrics and Blood Pressure

Body height and weight in units of 0.1 cm and 0.1 kg, respectively, were measured while wearing light cloths and barefoot at school. BMI was calculated as weight (kg)/height (m)^2^. BP was measured using an automated sphygmomanometer (HEM-707, HEM-757, or HEM-780, OMRON) after 5 min of sitting, before drawing blood. 

### 2.5. Analytic Variables

The students were categorized into five groups using gender-specific quintiles of added or free sugars intakes. In addition, dichotomous variables of added and free sugars intake were created based on whether each sugars intake was ≥5%E or not, and whether ≥10%E or not. A z-score of BMI (zBMI) was calculated using the LMS (Lambda-Mu-Sigma) method with the 1979–1981 Japanese reference [34]. Overweight, including obesity, was defined using BMI cutoffs of the International Obesity Task Force (IOTF) criteria, which corresponds to BMI of 25 kg/m^2^ at 18 years old [35], or as ≥1 standard deviation (SD) [36]. High risks were defined as LDL ≥ 120 mg/dL [37], HDL < 40 mg/dL, TG ≥ 150 mg/dL, Glu ≥ 100 mg/dL, and high BP (systolic BP ≥ 130 mmHg or diastolic BP ≥ 85 mmHg) [38]. Adolescent metabolic syndrome was defined as overweight instead of abdominal adiposity plus two risks of low HDL, high TG, high Glu, and high BP with the modified International Diabetes Federation (IDF) definition [38]. A continuous standardized variable of metabolic syndrome was calculated in accordance with a previous report [39] from the 6 following measurements used for the adolescent metabolic syndrome definition: zBMI, and the natural logarithmic transformed HDL, TG, Glu, SBP, and DBP (the z-score of HDL was inversed). As confounding factors, questions on sleeping duration (hr), number of siblings (1, 2, or ≥3), single parent, and physical activity (>2 times of exercise per week) were asked in the lifestyle questionnaire, and described in elsewhere [28]. Ages (in years) were calculated as (the blood drawing date−the birth date)/365.25. 

We excluded from the analysis the students with missing data, those who had diseases (heart disease, kidney disease, diabetes, hypertension), or implausible energy intake estimated from BDHQ (Figure 1). Plausible energy intake was ≥0.5 times the energy requirement at physical activity level 1, and ≤1.5 times the energy requirement at physical activity level 3, according to the Dietary Reference Intake for Japanese people [40]. In addition, we excluded the students who took breakfast before blood drawing or had missing records about taking breakfast.

### 2.6. Statistical Analysis

We statistically analyzed the data of 3242 students using SAS 9.4 (SAS Institute Japan Inc., Tokyo, Japan). Continuous and categorical variables were expressed as mean ± SD and as counts (%), respectively. Cardiometabolic risk factors (continuous variables), among the five ranked categories of sugars intake were calculated as the least square means with adjustment for age, gender, sleeping duration, number of siblings, single parent, physical activity, TEI, fat, total dietary fiber, and zBMI, and tested for trends as a simple linear relationship (*P_trend_*). When zBMI was a dependent variable, zBMI was excluded from confounders. Odds ratios (ORs) of high sugars intake (≥5%E, or ≥10%E) for dichotomous variables of metabolic markers were adjusted for age, gender, sleeping duration, number of siblings, single parent, physical activity, TEI, fat, total dietary fiber and zBMI. ORs for overweight were adjusted for all confounders excluding zBMI. For the sensitivity analysis, we analyzed the data of students who had a fasting blood test (*n* = 3242) with other definitions of metabolic syndrome. Since the prevalence of metabolic syndrome was low in this sample, high risks were defined as the highest 10% of the risk parameters in this sample [41]. We additionally analyzed the data separately by gender (*n* for males = 1659, and for females = 1583). Since the validity of the dietary data of these students might be higher than that of the parents of the students who responded instead [29], we analyzed the data of the students who responded to the BDHQ15y by themselves (*n* = 1626), excluding the data in which their parents responded to it instead of the students, or if helped by them (*n* = 1616). The statistical test results were considered significant when *p* values were < 0.05.

## 3. Results

Of the 3242 students who had a fasting blood test, 1659 (51.2%) were male. The students’ mean (SD) age was 13.56 ± 0.29 years for both sexes, and the mean (SD) BMI was 18.95 ± 2.70 kg/m^2^ for males and 19.54 ±2.68 kg/m^2^ for females (Table 1). The prevalence of overweight according to the IOTF was 154/1659 (9.3%) in males and 148/1583 (9.4%) in females, which were lower than those defined as ≥1 SD (16.0% (265/1659) in males and 19.4% (307/1583) in females). The prevalence of metabolic syndrome was higher when using the ≥1 SD definition for overweight than when using the IOTF definition. 

Intakes of added and free sugars were as follows: males, 7.6 ± 4.0%E, and 8.4 ± 4.4%E; females, 7.9 ± 4.1%E, and 8.8 ± 4.6%E, respectively. Males who had high intakes of added sugars ≥5%E or ≥10%E accounted for 72.9% or 20.6% of the population, and females accounted for 78.0% or 22.1%, respectively. Males with high intake of free sugars ≥5%E or ≥10%E accounted for 78.7% or 27.1% of the population and females accounted for 82.4% or 29.9%, respectively.

Categories based on quintiles of sugars intake were positively associated with Glu level and z-score of metabolic syndrome with adjustment for confounders including TEI, and zBMI (*P_trend_* < 0.001, and *P_trend_* ≤ 0.02, respectively; Table 2); differences between the highest and lowest intakes were 1.0–1.3 mg/dL in Glu, and 0.06–0.12 in z-score of metabolic syndrome. Categorized added and free sugars intakes were positively associated with systolic BP level (*P_trend_* ≤ 0.025, 1.7–2.3 mmHg). 

High intakes of added or free sugars were not significantly associated with overweight or other risks, except for added sugars intake ≥10%E for Glu ≥ 100 mg/dL (OR 1.51, 95% confidence interval (CI) 1.04–2.19, *p* = 0.031; Table 3). For the sensitivity analysis, when high risks were defined as the highest 10% of risk levels, only ORs of high added sugars ≥5%E and ≥10%E and free sugars ≥10%E with Glu were significant (*n* = 3242; *p* ≤ 0.040). Additionally, the OR of added sugar ≥5%E with high LDL and the ORs of free sugars ≥5%E with high LDL and high SBP were significant (*p* = 0.017–0.044); but the ORs for metabolic syndrome were insignificant. When analyzing the data separately by gender (*n* for males = 1659, and for females = 1583), similar ORs were obtained (data not shown). The association between added sugars intake ≥10%E and high Glu retained their significance even when the analysis was restricted to the students who responded to the BDHQ by themselves (*n* = 1626; OR 1.49; 95% CI 1.03–2.15, *p* = 0.033). 

## 4. Discussion

Ranked variables of added and free sugars intake in Japanese secondary school students were associated with their plasma glucose level, BP level, and metabolic syndrome score as continuous variables. When using dichotomous variables, high intake of added sugars ≥10%E was also significantly associated with high plasma glucose ≥100 mg/dL, but other associations of high sugars intake were not significant, even when using added or free sugars intake, and using overweight prevalence based on two different definitions.

Reports on sugars intake in Japanese adolescents are scarce. Dietary records of 915 Japanese primary and secondary school students aged 8–14 years revealed that students consumed 51.7–52.5% of energy from carbohydrates, which included total sugars (12.3–12.8%E) and free sugars (5.8–6.0%E) [32]. The sugars intake estimated from dietary records is lower than the sugars intake in this current study in which students were aged 13–14 years. Even though Japanese adolescents could consume more sugars than children, mean added or free sugars intake of Japanese secondary school students did not exceed 10%E, unlike those of European and American adolescents [18,42,43]. 

Higher consumption of SSBs was associated with higher fasting glucose in the UK adolescents [44] and high homeostasis model assessment-insulin resistance (HOMA-IR) in the US adolescents [45]. In a cross-sectional study of the US adolescents aged 12–19 years, however, added sugars intake was not significantly associated with glucose, insulin, or HOMA-IR [46]. Although we did not measure insulin levels and HOMA-IR, we demonstrated the association between added or free sugars and fasting glucose level. The associations were not sufficient to identify any adolescent with a high risk because only the OR of high added sugars intake ≥10%E was significant. 

A possible mechanism of the association between high sugars intake and high BP is the elevation of uric acid intermediates. Fructose consumption induces purine degradation [47], and a slight elevation of blood glucose (within the normal range) enhances uric acid reabsorption [48]. Although uric acid levels were not measured in this study, previous cross-sectional studies showed that intake of SSBs and sugars was significantly associated with uric acid level in the US adolescents [46,49]. However, the results of the association between sugars intake and BP are inconsistent. The association of the z-score of SBP and SSBs in studies using data from the US National Health and Nutrition Examination Survey was significant, while studies using the continuous variable of BP with SSBs [45] or studies on high BP with added sugars [46] showed a non-significant association. In our study, the continuous variable of BP was significantly associated with sugars intake, but the ORs for high BP were not significant. 

Like other risks, metabolic syndrome score as a continuous variable was significantly associated with sugars intake in this study; but the ORs were not significant. The definition of metabolic syndrome proposed by the IDF includes central adiposity instead of total adiposity BMI as an essential item. There is controversy regarding whether BMI, or central adiposity is a stronger predictor of cardiovascular risks [50]. We did not measure the students’ waist circumference, but we determined the associations using prevalence based on the two definitions of overweight, IOTF and ≥1 SD, because Asians have less BMI, but similar associations with cardiovascular risks [15,51]. Several studies on Asian adults showed that BMI is associated with diabetes, hypertension, metabolic syndrome, and central obesity indices [50,51,52,53]. Hence, the results of this study indicate that the association between sugars intake and metabolic syndrome may be weak in Japanese adolescents. 

Another possible explanation for low ORs of sugars intake for metabolic syndrome is the difference in the ranges of sugars intake between Japan and European or American countries [18,42,43]. Adolescents in this study consumed 7.6–7.9%E of added sugars and 8.4–8.8%E of free sugars. Children and adolescents in Portugal [54], Spain [55], Slovenia [56], Colombia, and Ecuador [43] had a mean intake of 9.5–11.6%E and 9.8–10.1%E, respectively. The intake in this study was lower than that in previous studies; in particular in the Netherlands [57], the UK [58], the US [42], Argentina, Brazil, Chile, Costa Rica, Peru, and Venezuela [43] (12.9–18.4%E and 17.6–20.7%E, respectively). Most literatures that reported significant associations of cardiometabolic risks came from the US [45,46,49], where the prevalence of metabolic syndrome based on the IDF definition, i.e., 5.2–9.5% at age 14 years [59,60] was higher than that in the current study. In the population with low intake and low prevalence, we could not find significant ORs. 

Previous studies on Asian adolescents did not show significant associations between intake of sugars or SSBs and adiposity [23,24,61]. Since sugars intake is required to gain weight from excess energy intake [7], the association between sugars intake and obesity may be confounded by energy intake. Adjusting for confounding factors is too complicated. A cross-sectional study evaluating any association with obesity is prone to biases in an unexpected direction. In order to examine the associations with adiposity, a prospective cohort study or a randomized control trial should be implemented [1,62].

A large sample size, one of the strengths of this study, revealed the associations of added or free sugars intake with Glu, SBP, and metabolic syndrome score. However the ORs for metabolic syndrome or most of other risks were not significant even with a large sample size, which is comparable to those of other studies on Asian adolescents [24,25]. Other than BMI, we used metabolic variables that might be a rare complicated confounder of energy intake. Specifically, we analyzed the data while restricting the data to fasting blood samples, even though plasma glucose levels returned to normal after only 2 h in healthy adolescents. In this sample, the mean plasma glucose for students with and without breakfast was 90.03 mg/dL and 90.16 mg/dL, respectively; this is the second strength of this study. However, one limitation in this study was that BDHQ was a self-administered questionnaire. The students may respond to the questionnaire with knowledge about a healthy lifestyle. This desirability bias might weaken the true association. Estimation of sugars intake from the BDHQ might be a second limitation. Added and free sugars intake had a plausible range in Japanese participants, but the results warrant further studies with a different sample and other dietary assessment. 

## 5. Conclusions

Categorized intake of added and free sugars was associated with the continuous variables of cardiovascular risk levels in Japanese adolescents whose mean intake was less than 10%E, after adjustment for confounders including energy intake and zBMI. When using dichotomous variables, however, we are unable to identify adolescents with a high multiple risk based on high sugars intake. This is because we did not find significant associations except that between high added sugars intake and high fasting glucose. In particular, we could not find significant ORs for metabolic syndrome under any situation, such as in stratified analysis by gender, using different definitions of metabolic syndrome, or in a restricted sample. We recommend a reduction in the amount of added or free sugars in Japanese adolescents with high intake levels. For public health concerns, recommendation of added sugars intake <10%E should be considered to prevent glucose intolerance in adolescents. However, further recommendation to lower added sugars intake <5%E was not supported by this study’s findings.

## Figures and Tables

**Figure 1 nutrients-12-02046-f001:**
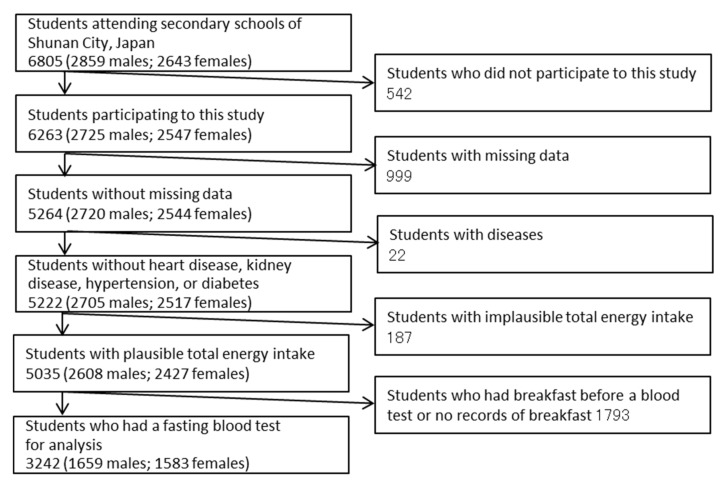
Subjects’ selection.

**Table 1 nutrients-12-02046-t001:** Characteristics of the study participants.

	Males, *n* = 1659	Females, *n* = 1583
Age, years	13.56 ± 0.29	13.56 ± 0.29
BMI, kg/m^2^	19.0 ± 2.7	19.5 ± 2.7
zBMI	0.01 ± 1.06	0.13 ± 1.07
Total energy, kcal/day	2414 ± 655	2014 ± 538
Carbohydrate, %E	55.3 ± 6.7	53.2 ± 6.4
Added sugars, %E	7.6 ± 4.0	7.9 ± 4.1
Free sugars, %E	8.4 ± 4.4	8.8 ± 4.6
Overweight (IOTF)	154 (9.3)	148 (9.4)
Overweight (≥1 SD)	265 (16.0)	307 (19.4)
High LDL (≥120 mg/dL)	122 (7.4)	166 (10.5)
Low HDL (<40 mg/dL)	20 (1.2)	8 (0.5)
High TG (≥150 mg/dL)	34 (2.1)	27 (1.7)
High Glu (≥100 mg/dL)	106 (6.4)	47 (3.0)
High SBP (≥130 mmHg)	211 (12.7)	92 (5.8)
High DBP (≥185 mmHg)	48 (2.9)	60 (3.8)
High BP (high SBP, or high DBP)	232 (14.0)	134 (8.5)
MS (IOTF)	14 (0.8)	6 (0.4)
MS (≥1 SD)	17 (1.0)	9 (0.6)

Data are presented as mean ± SD or counts (%). BMI, body mass index; zBMI, z-score of BMI; %E, percentage of total energy intake; IOTF, International Obesity Task Force; SD, standard deviation; LDL, low-density lipoprotein cholesterol; HDL, high-density lipoprotein cholesterol; TG, triglyceride; Glu, fasting glucose, SBP and DBP, systolic and diastolic blood pressure; MS, metabolic syndrome. Metabolic syndrome was determined as overweight (IOTF, or ≥1 SD) plus two or more risks of low HDL, high TG, high fasting glucose, and high BP.

**Table 2 nutrients-12-02046-t002:** Least square means of risk parameters according to quintile categories of sugars intake.

	Q1	Q2	Q3	Q4	Q5	*P_trend_*
**Added sugars, *n***	647	649	650	648	648	
**%E**	3.36 ± 0.83	5.37 ± 0.49	6.99 ± 0.53	8.89 ± 0.71	14.01 ± 3.87	
zBMI	0.11	0.10	0.03	0.01	0.08	0.289
LDL, mg/dL	88.8	90.1	89.8	89.8	90.2	0.393
HDL, mg/dL	67.3	67.7	68.4	67.7	67.2	0.883
TG, mg/dL	58.3	59.1	57.2	59.6	59.1	0.587
Glu, mg/dL	89.5	89.7	89.9	90.5	90.8	**<0.001**
SBP, mmHg	112.7	114.4	114.1	114.9	115.0	**<0.001**
DBP, mmHg	67.8	68.2	67.8	68.3	68.5	0.152
zMS	−0.06	0.00	−0.04	0.04	0.06	**<0.001**
**Free sugars, *n***	647	649	649	649	648	
**%E**	3.73 ± 0.96	5.97 ± 0.56	7.78 ± 0.59	9.9 ± 0.76	15.58 ± 4.28	
zBMI	0.13	0.07	0.05	−0.01	0.09	0.277
LDL, mg/dL	89.0	88.9	90.1	90.1	90.6	0.113
HDL, mg/dL	67.2	68.2	68.3	67.7	67.0	0.513
TG, mg/dL	58.7	57.3	58.4	60.9	58.1	0.532
Glu, mg/dL	89.6	89.5	90.2	90.3	90.8	**<0.001**
SBP, mmHg	113.0	114.7	114.0	114.7	114.7	**0.025**
DBP, mmHg	67.7	68.6	67.9	67.8	68.6	0.404
zMS	−0.05	−0.01	−0.02	0.02	0.05	**0.001**

Data are presented as mean intake ± standard deviation. %E, percentage of total energy intake; zBMI, z-score of body mass index LDL, low-density lipoprotein cholesterol; HDL, high-density lipoprotein cholesterol; TG, triglyceride; Glu; fasting glucose; SBP, and DBP, systolic and diastolic blood pressure; zMS, z-score of metabolic syndrome is calculated as the mean of the z-scores of BMI, HDL (inversed), TG, Glu, SBP, and DBP. Least square means were adjusted for age, gender, sleeping duration, number of siblings, single parent, physical activity, total energy, fat, and total dietary fiber. Parameters other than zBMI were additionally adjusted for zBMI. *P_trend_* indicates the tests of simple linear trends of the risk levels among the ranked categories of sugars intake.

**Table 3 nutrients-12-02046-t003:** Odds ratios of high sugars intake for overweight, high LDL, TG, Glu, SBP, DBP or BP, low HDL, and MS.

	Added Sugars Intake					Free Sugars Intake				
	Low (ref.)	High	OR	(95% CI)	*p*	Low (ref.)	High	OR	(95% CI)	*p*
**Cutoff, 5%E**	822	2420				632	2610			
Overweight (IOTF)	86 (10.5)	216 (8.9)	0.95	(0.72, 1.26)	0.744	67 (10.6)	235 (9.0)	0.97	(0.71, 1.31)	0.819
Overweight (≥1 SD)	154 (18.7)	418 (17.3)	0.98	(0.79, 1.21)	0.841	119 (18.8)	453 (17.4)	0.98	(0.78, 1.24)	0.874
High LDL	59 (7.2)	229 (9.5)	1.29	(0.94, 1.76)	0.112	44 (7.0)	244 (9.3)	1.32	(0.94, 1.87)	0.113
Low HDL	6 (0.7)	22 (0.9)	1.38	(0.53, 3.58)	0.511	5 (0.8)	23 (0.9)	1.26	(0.46, 3.5)	0.653
High TG	21 (2.6)	40 (1.7)	0.79	(0.45, 1.41)	0.429	17 (2.7)	44 (1.7)	0.78	(0.43, 1.43)	0.419
High Glu	32 (3.9)	121 (5.0)	1.25	(0.83, 1.90)	0.291	27 (4.3)	126 (4.8)	1.11	(0.71, 1.73)	0.641
High SBP	77 (9.4)	226 (9.3)	1.20	(0.89, 1.60)	0.233	60 (9.5)	243 (9.3)	1.20	(0.87, 1.65)	0.262
High DBP	26 (3.2)	82 (3.4)	1.10	(0.69, 1.75)	0.704	17 (2.7)	91 (3.5)	1.36	(0.79, 2.34)	0.273
High BP	92 (11.2)	274 (11.3)	1.15	(0.88, 1.51)	0.300	71 (11.2)	295 (11.3)	1.16	(0.87, 1.56)	0.309
MS (IOTF)	6 (0.7)	14 (0.6)	1.24	(0.36, 4.28)	0.738	5 (0.8)	15 (0.6)	1.22	(0.34, 4.41)	0.765
MS (≥1 SD)	6 (0.7)	20 (0.8)	1.53	(0.51, 4.60)	0.452	5 (0.8)	21 (0.8)	1.53	(0.47, 4.94)	0.481
**Cutoff, 10%E**	2551	691				2319	923			
Overweight (IOTF)	237 (9.3)	65 (9.4)	1.05	(0.78, 1.41)	0.763	216 (9.3)	86 (9.3)	1.03	(0.79, 1.35)	0.830
Overweight (≥1 SD)	452 (17.7)	120 (17.4)	0.99	(0.79, 1.24)	0.918	417 (18)	155 (16.8)	0.93	(0.75, 1.14)	0.460
High LDL	213 (8.3)	75 (10.9)	1.26	(0.95, 1.68)	0.113	195 (8.4)	93 (10.1)	1.17	(0.89, 1.52)	0.263
Low HDL	22 (0.9)	6 (0.9)	1.04	(0.41, 2.63)	0.942	21 (0.9)	7 (0.8)	0.86	(0.36, 2.08)	0.744
High TG	54 (2.1)	7 (1.0)	0.50	(0.22, 1.12)	0.093	51 (2.2)	10 (1.1)	0.53	(0.26, 1.06)	0.071
High Glu	109 (4.3)	44 (6.4)	1.51	(1.04, 2.19)	**0.031**	102 (4.4)	51 (5.5)	1.28	(0.90, 1.82)	0.171
High SBP	243 (9.5)	60 (8.7)	0.94	(0.69, 1.28)	0.683	222 (9.6)	81 (8.8)	0.97	(0.73, 1.28)	0.826
High DBP	88 (3.4)	20 (2.9)	0.85	(0.51, 1.41)	0.529	77 (3.3)	31 (3.4)	1.03	(0.67, 1.59)	0.891
High BP	293 (11.5)	73 (10.6)	0.94	(0.70, 1.24)	0.642	265 (11.4)	101 (10.9)	1.00	(0.78, 1.29)	0.999
MS (IOTF)	18 (0.7)	2 (0.3)	0.61	(0.12, 3.12)	0.551	17 (0.7)	3 (0.3)	0.63	(0.15, 2.63)	0.528
MS (≥1 SD)	22 (0.9)	4 (0.6)	0.95	(0.29, 3.07)	0.930	20 (0.9)	6 (0.7)	1.04	(0.37, 2.93)	0.939

Data are presented as *n* (%). %E, percentage of total energy intake; IOTF, International Obesity Task Force; SD, standard deviation; LDL, low-density lipoprotein cholesterol; HDL, high-density lipoprotein cholesterol; TG, triglyceride; Glu, fasting glucose; BP, blood pressure; MS, metabolic syndrome. MS was determined as overweight (IOTF or ≥1 SD) plus two or more risks of low HDL, high TG, high Glu, and high BP; OR, odds ratio; CI, confidence interval. ORs were adjusted for age, gender, sleeping duration, number of siblings, single parent, physical activity, total energy, fat, and total dietary fiber. ORs, for cardiometabolic risks other than overweight, were additionally adjusted for a z-score of BMI.

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
