# Peer review of "Added and Free Sugars Intake and Metabolic Biomarkers in Japanese Adolescents"

_nutrients, 2020, doi:10.3390/nu12072046_

Round 1
Reviewer 1 Report
Added and free sugars consumption influences the development of cardiometabolic diseases. Therefore, different public health institutions recommend limiting free sugars consumption to less than 10% or 5% of total energy intake. The authors aimed to further explore possible correlation between added and free sugars intake and metabolic biomarkers in Japanese student population using brief-type self-administered diet history questionnaire. The results showed that added sugar above 10% of total energy intake was associated with higher fasting plasma glucose, blood pressure, and metabolic syndrome score. The study also showed, that Japanese students consume less added or free sugar compared to their American and European peers, and have lower levels of obesity.
The research field of added and free sugars consumption is an important one given the large mismatch between nutritional guidelines and eating patterns in developed as well as developing countries. Japan has the lowest sugar consumption and prevalence of obesity among developed countries, but still has relatively high rates of cardiovascular disease. The manuscript provides an interesting new piece of research within the field, but could be improved further.
Major revisions:
- Throughout the entire article, including the discussion, it is not entirely clear, which findings were significantly associated with added and free sugar and which were not. For example, in the discussion it says: “Added and free sugar intake in Japanese secondary school students was associated with their plasma glucose level, blood pressure, and metabolic syndrome score. Except for the association between added sugar intake >10%E mg/dL, however, high sugar intake was poorly associated with high cardiovascular risks, even when using added or free sugar intake...”, which is already contradicting, and not clear, what kind of association was observed with blood pressure and metabolic syndrome score.
- From the article is also not entirely clear, which analyses included all participants, and which excluded those, whose questionnaires were filled-in by the parents.
- In the statistical analysis it needs to be written which p value was considered significant. You seem to use different p-value cut-off points for different analyses.
Minor revisions:
- Whole paper: Consistently with the WHO nomenclature, added and free sugars should be written in plural as “sugars” instead of “sugar”
- The abstract would benefit from the clearer conclusion (last sentence).
- English language can be further improved.
Author Response
Thank you very much for your reviewing out article, which helps us to improve the manuscript.
Major revisions:
- Throughout the entire article, including the discussion, it is not entirely clear, which findings were significantly associated with added and free sugar and which were not. For example, in the discussion it says: “Added and free sugar intake in Japanese secondary school students was associated with their plasma glucose level, blood pressure, and metabolic syndrome score. Except for the association between added sugar intake >10%E mg/dL, however, high sugar intake was poorly associated with high cardiovascular risks, even when using added or free sugar intake...”, which is already contradicting, and not clear, what kind of association was observed with blood pressure and metabolic syndrome score.
We have revised the Abstract and Conclusion. We clarified the associations between ranked intake and continuous variable, and those between dichotomous and dichotomous variables. Dichotomous variables lose some information, but have clinical meanings.
Abstract. (Lines 25–27)
“Association was observed between added or free sugars intake and cardiometabolic biomarkers in Japanese adolescents, and added sugars intake <10%E could prevent glucose intolerance, but not metabolic syndrome.”
Discussion (Lines 197–202)
“Ranked variables of added and free sugars intake in Japanese secondary school students were associated with their plasma glucose level, BP level, and metabolic syndrome score as continuous variables. When using dichotomous variables, high intake of added sugars ≥10%E was also significantly associated with high plasma glucose ≥100 mg/dL, but other associations of high sugars intake were not significant, even when using added or free sugars intake, and using overweight prevalence based on two different definitions.”
Conclusion (Lines 267–274)
“Categorized intake of added and free sugars was associated with the continuous variables of cardiovascular risk levels in Japanese adolescents whose mean intake was less than 10%E, after adjustment for confounders including energy intake and zBMI. When using dichotomous variables, however, we are unable to identify adolescents with a high multiple risk based on high sugars intake. This is because we did not find significant associations except that between high added sugars intake and high fasting glucose. In particular, we could not find significant ORs for metabolic syndrome under any situation, such as in stratified analysis by gender, using different definitions of metabolic syndrome, or in a restricted sample.”
- From the article is also not entirely clear, which analyses included all participants, and which excluded those, whose questionnaires were filled-in by the parents.
We have revised the Methods and Result about the sensitivity analysis to clarify sample sizes
Methods (Lines 135–142)
“For the sensitivity analysis, we analyzed the data of students who had a fasting blood test (N = 3242) with other definitions of metabolic syndrome. Since the prevalence of metabolic syndrome was low in this sample, high risks were defined as the highest 10% of the risk parameters in this sample [41]. We additionally analyzed the data separately by gender (N for males = 1659, and for females = 1583). Since the validity of the dietary data of these students might be higher than that of the parents of the students who responded instead [29], we analyzed the data of the students who responded to the BDHQ15y by themselves (N = 1626), excluding the data in which their parents responded to it; instead of the students, or if helped by them (N = 1616).”
Results (Lines 173–181)
“For the sensitivity analysis, when high risks were defined as the highest 10% of risk levels, only ORs of high added sugars ≥5%E and ≥10%E and free sugars ≥10%E with Glu were significant (N = 3242; P ≤ 0.040). Additionally, the OR of added sugar ≥5%E with high LDL and the ORs of free sugars ≥5%E with high LDL and high SBP were significant (P = 0.017–0.044); but the ORs for metabolic syndrome were insignificant. When analyzing the data separately by gender (N for males = 1659, and for females = 1583), similar OR were obtained (data not shown). The association between added sugars intake ≥10%E and high Glu retained their significance even when the analysis was restricted to students who responded to the BDHQ by themselves (N = 1626; OR 1.49; 95% CI 1.03–2.15, P = 0.033).”
- In the statistical analysis it needs to be written which p value was considered significant. You seem to use different p-value cut-off points for different analyses.
We have added a sentence about a significance level. (Lines 142–143)
“The statistical test results were considered significant when p values were <0.05.”
Minor revisions:
- Whole paper: Consistently with the WHO nomenclature, added and free sugars should be written in plural as “sugars” instead of “sugar”
We have corrected them throughout the manuscript.
- The abstract would benefit from the clearer conclusion (last sentence).
We have revised the Abstract. (Lines 25–27)
“Association was observed between added or free sugars intake and cardiometabolic biomarkers in Japanese adolescents, and added sugars intake <10%E could prevent glucose intolerance, but not metabolic syndrome.”
- English language can be further improved.
The text was English-edited as seen in the attached certificate.

Reviewer 2 Report
A very interesting work of Okuda and colleagues, focusing on added and free sugar intake in Japanese adolescents and their association with anthropometric parameters as well as parameters of glycemic control and lipid metabolism. Although a very nice approach, the following points have to be mentioned:
- In abstract, the last sentence (conclusion) does not really fit to the authors conclusion in line 226/227 and the interpretation of the findings. Please highlight the weak or rather non-significant association of dietary sugar intake and MS in Japanese adolescents to the abstract and replace it with the last current last sentence.
- In line 41, a paragraph should be added explaining that the effect of dietary sugars on body weight and with that diet-related diseases are mediated by increased energy intake rather than sugars per se, as demonstrated by the findings of Te Morenga et al. (already cited as reference 1) and Fattore et al. 2017 Am J Clin Nutr (both systematic reviews and meta-analyses with the highest scientific evidence).
- In results and discussion, the association between added sugar intake and blood glucose level as well as blood pressure should be adjusted for important confounders, including BMI as well as energy intake. A higher body weight and a higher energy intake can have an important impact on the current findings. If not possible, these confounders should be discussed extensively in the discussion, e.g. limitations.
- In conclusion, the authors write that added and free sugar intake is associated with cardiovascular risks. However, this statement does not reflect the findings of the study. Dietary sugar intake was not associated with anthropometric parameters, blood lipids or blood pressure. The authors only find an association with elevated blood glucose level without adjustment for energy intake or BMI. Therefore, the authors should conclude that besides blood glucose level no significant associations were found with the investigated parameters of this study.
Author Response
Thank you for your review comments, which help us to improve our manuscripts.
- In abstract, the last sentence (conclusion) does not really fit to the authors conclusion in line 226/227 and the interpretation of the findings. Please highlight the weak or rather non-significant association of dietary sugar intake and MS in Japanese adolescents to the abstract and replace it with the last current last sentence.
We have revised the last sentence of the Abstract to describe significant and insignificant results (Lines 25–27)
“Association was observed between added or free sugars intake and cardiometabolic biomarkers in Japanese adolescents, and added sugars intake <10%E could prevent glucose intolerance, but not metabolic syndrome.”
- In line 41, a paragraph should be added explaining that the effect of dietary sugars on body weight and with that diet-related diseases are mediated by increased energy intake rather than sugars per se, as demonstrated by the findings of Te Morenga et al. (already cited as reference 1) and Fattore et al. 2017 Am J Clin Nutr (both systematic reviews and meta-analyses with the highest scientific evidence).
Thank you for your comments. We have added a sentence in the Introduction and a reference (Lines 64–68; Reference 26).
“In meta-analyses on energy-control trials mostly from Europe and the USA [1, 26], observed effects of sugars intake on body weight, blood pressure (BP), and blood lipid profile are considered via an excess of energy intake in conjunction with an excess of sugars intake; so we examined the associations with adjustment for TEI. “
- In results and discussion, the association between added sugar intake and blood glucose level as well as blood pressure should be adjusted for important confounders, including BMI as well as energy intake. A higher body weight and a higher energy intake can have an important impact on the current findings. If not possible, these confounders should be discussed extensively in the discussion, e.g. limitations.
The association between added sugar intake and blood glucose level as well as blood pressure was already adjusted for BMI, and energy intake. We have revised the Methods and added each item of confounding factors to clarify them.
Methods (Lines 128–135)
“Cardiometabolic risk factors (continuous variables), among the five ranked categories of sugars intake were calculated as the least square means with adjustment for age, gender, sleeping duration, number of siblings, single parent, physical activity, TEI, fat, total dietary fiber, and zBMI, and tested for trends as a simple linear relationship (Ptrend). When zBMI was a dependent variable, zBMI was excluded from confounders. Odds ratios (ORs) of high sugars intake (≥5%E, or ≥10%E) for dichotomous variables of metabolic markers were adjusted for age, gender, sleeping duration, number of siblings, single parent, physical activity, TEI, fat, total dietary fiber and zBMI. ORs for overweight were adjusted for all confounders excluding zBMI.”
Results (Lines 165–166)
“Categories based on quintiles of sugars intake was positively associated with Glu level and z-score of metabolic syndrome with adjustment for confounders including TEI, and zBMI ..“
Conclusion (Lines 267–269)
“Categorized intake of added and free sugars was associated with the continuous variables of cardiovascular risk levels in Japanese adolescents whose mean intake was less than 10%E, after adjustment for confounders including energy intake and zBMI.”
- In conclusion, the authors write that added and free sugar intake is associated with cardiovascular risks. However, this statement does not reflect the findings of the study. Dietary sugar intake was not associated with anthropometric parameters, blood lipids or blood pressure. The authors only find an association with elevated blood glucose level without adjustment for energy intake or BMI. Therefore, the authors should conclude that besides blood glucose level no significant associations were found with the investigated parameters of this study.
We found significant associations for dichotomous LDL, and BP in the sensitivity analysis using definition >10% high risks, so we revised the Conclusion to confine an insignificant result to MS. (Lines 269–274)
“When using dichotomous variables, however, we are unable to identify adolescents with a high multiple risk based on high sugars intake. This is because we did not find significant associations except that between high added sugars intake and high fasting glucose. In particular, we could not find significant ORs for metabolic syndrome under any situation, such as in stratified analysis by gender, using different definitions of metabolic syndrome, or in a restricted sample.”

Round 2
Reviewer 1 Report
Thank you for improving this manuscript considerably. Reading the paper once again I just got another suggestion for improving the discussion. Your study also report consumption of free sugar in adolescents, and it would be very useful if reported values are compared with a few available studies conducted outside Japan. You already provide so introduction on this topic in the introduction, but this could be also mentioned in the discussion section, in relationship to you results. In the discussion section you currently mention (L238-245) that “even though Japanese adolescents could consume more sugars than children, mean added or free sugars intake of Japanese secondary school students did not exceed 10%E, unlike those of European and American adolescents [18, 41, 42]”. It would be also useful to compare %TEI levels, where Japanese results are not so much different than in some other countries. For example, you report %TEI of free sugar levels are somehow comparable with Spain (https://www.mdpi.com/2072-6643/9/3/275 ), Slovenia (https://www.mdpi.com/2072-6643/12/6/1729) and Portugal (http://dx.doi.org/10.1017/S1368980019002519), but quite lower than UK (https://www.mdpi.com/2072-6643/12/2/393) and Netherlands (https://www.mdpi.com/2072-6643/8/2/70). This can also partially explain the differences in incidence of non-communicative disease across different countries.
Please also add more details on the ethical approval of the study (particularly date of the approval.
Author Response
Response to the comments of the Reviewer 1
Thank you for your suggestion, which helped us to add worthwhile discussion. We have corrected the texts on the Review’s advice.
Comment1: Your study also report consumption of free sugar in adolescents, and it would be very useful if reported values are compared with a few available studies conducted outside Japan. You already provide so introduction on this topic in the introduction, but this could be also mentioned in the discussion section, in relationship to you results. In the discussion section you currently mention (L238-245) that “even though Japanese adolescents could consume more sugars than children, mean added or free sugars intake of Japanese secondary school students did not exceed 10%E, unlike those of European and American adolescents [18, 41, 42]”. It would be also useful to compare %TEI levels, where Japanese results are not so much different than in some other countries. For example, you report %TEI of free sugar levels are somehow comparable with
Spain (https://www.mdpi.com/2072-6643/9/3/275 ),
Slovenia (https://www.mdpi.com/2072-6643/12/6/1729) and
Portugal (http://dx.doi.org/10.1017/S1368980019002519),
but quite lower than
UK (https://www.mdpi.com/2072-6643/12/2/393) and
Netherlands (https://www.mdpi.com/2072-6643/8/2/70). This can also partially explain the differences in incidence of non-communicative disease across different countries.
Response 1: Thank you for your suggestion; it is important to interpret the results. We have added a paragraph citing suggested references. Lines 245–255, and Ref. 54-60.
Discussion, Lines 245–255.
“Another possible explanation for low ORs of sugars intake for metabolic syndrome is the difference of the ranges of sugars intake between Japan and European or American countries [18, 42, 43]. Adolescents in this study consumed 7.6–7.9%E of added sugars and 8.4–8.8%E of free sugars. Children and adolescents in Portugal [54], Spain [55], Slovenia [56], Colombia, and Ecuador [43] had a mean intake of 9.5–11.6%E and 9.8–10.1%E, respectively. The intake in this study was lower than that in previous studies; in particular in the Netherlands [57], the UK [58], the US [42] , Argentina, Brazil, Chile, Costa Rica, Peru, and Venezuela [43] (12.9–18.4%E and 17.6-20.7%E, respectively). Most literatures that reported significant associations of cardiometabolic risks came from the US [45, 46, 49], where the prevalence of metabolic syndrome based on the IDF definition, i.e., 5.2–9.5% at age 14 years [59, 60] was higher than that in the current study. In the population with low intake and low prevalence, we could not find significant ORs.”
Comment 2: Please also add more details on the ethical approval of the study (particularly date of the approval.
Response 2: We have added informations. Lines 75–78.
“This study protocol was in accordance the guidelines laid down in with the Declaration of Helsinki, and approved by the Ethics Committee of Yamaguchi University Hospital (the approval number H17-14 on May 18, 2005, H17-14-2 on March 22, 2006, H22-158 on January 26, 2011, and H22-158-[1] on March 22, 2017) and the education board of Shunan City.”
